# Comparing the External Loads Encountered during Competition between Elite, Junior Male and Female Basketball Players

**DOI:** 10.3390/ijerph17041456

**Published:** 2020-02-24

**Authors:** Rubén Portes, Sergio L. Jiménez, Rafael M. Navarro, Aaron T. Scanlan, Miguel-Ángel Gómez

**Affiliations:** 1Faculty of Sport Sciences, Universidad Europea De Madrid, 28670 Madrid, Spain; rubenportes@gmail.com (R.P.); rafaelmanuel.navarro@universidadeuropea.es (R.M.N.); 2Human Exercise and Training Laboratory, School of Health, Medical and Applied Sciences, Central Queensland University, Rockhampton 4701, Australia; A.Scanlan@cqu.edu.au; 3Faculty of Physical Activity and Sport Sciences, Technical University of Madrid, 28040 Madrid, Spain; miguelangel.gomez.ruano@upm.es

**Keywords:** sex, gender, team sports, accelerometer, local positioning system

## Abstract

The aim of the present study was to compare external loads (EL) between elite, junior, male and female basketball players. Male (n = 25) and female players (n = 48) were monitored during 11 competitive matches (3 matches per team). EL was measured using local positioning system and microsensor technology to determine total, high-intensity (14–21 km·h^−1^), and sprint (>21 km·h^−1^) distance (m) covered, total (n) and relative (n·min^−1^) accelerations and decelerations, ratio of accelerations:decelerations, and total (arbitrary units [AU]) and relative (AU·min^−1^) player load. EL was compared between sexes overall and according to each playing position (guards, forwards, and centers). Males covered larger (*p* < 0.05) high-intensity and sprint distances, and completed more (*p* < 0.05) decelerations than females; while female players experienced a greater (*p* < 0.05) ratio of accelerations:decelerations. Greater decelerations (*p* < 0.05) were observed for males in the guard position compared to females, while more (*p* < 0.05) accelerations·min^−1^ were apparent for females in the forward position compared to males. The current findings indicate differences in EL, particularly the high-intensity and acceleratory demands, exist between elite, junior, male and female basketball players during competition and are affected by playing position. These outcomes can be used in developing sex- and position-specific training plans, and in turn improving the physical preparedness of junior basketball players for competition demands at the elite level.

## 1. Introduction

Monitoring of external load (EL) in basketball is essential to understand the physical demands experienced by players during training and competition [1]. EL is defined as the work completed by athletes, measured independently of their internal (e.g., heart rate, rating of perceived exertion) responses [2]. Time motion analyses, local positioning systems (LPS), and accelerometers are commonly used to quantify EL in basketball players during competition, with an increasing number of studies emerging in this area [3,4,5,6,7,8].

While the available research reporting on EL in basketball players is growing, much of the existing data are indicative of male players, with female players being largely under-represented [9,10,11,12,13]. In this regard, it is difficult to develop a definitive consensus on key differences in EL between male and female basketball players, especially given only one study has directly compared EL between sexes during basketball competition [11]. Specifically, Scanlan et al. [11] compared the match activities of semi-professional, adult, male and female players competing in separate competitions using video-based time-motion analysis. The authors reported some distinct differences between sexes with female backcourt (guard positions) players performing more (*p* < 0.05) low-intensity shuffling (1.2 ± 0.0 vs 0.9 ± 0.1 n·min^−1^) and jumping (1.1 ± 0.1 vs 0.8 ± 0.1 n·min^−1^) and spending longer (*p* < 0.05) jogging (21.8 ± 0.7 vs 19.9 ± 0.9 s·min^−1^) than male backcourt players. In contrast, female frontcourt (forward and center positions) players performed more (*p* < 0.05) upper-body activity (5.4 ± 0.4 vs 4.4 ± 0.3 n·min^−1^) than male frontcourt players. Overall, female players covered greater (*p* < 0.05) distances running (45.7 ± 1.4 vs 42.1 ± 1.7 m·min^−1^) than male players, while male players spent longer (3.0 ± 0.2 vs 2.5 ± 0.3 s·min^−1^, *p* < 0.05) and travelled significantly larger distances (9.7 ± 0.7 vs 8.4 ± 0.3 m·min^−1^, *p* < 0.05) dribbling than female players. While these data provide some important foundation insights regarding potential differences in EL between male and female players during actual competition, more research is required to develop a definitive understanding on this topic. 

Existing comparative EL data between sexes are indicative of sub-elite, adult players, and given the variations in match demands that exist across playing levels [3] as well as the morphological and physiological differences reported between junior and senior players [14], existing outcomes may not be transferable to elite, junior players. Data representative of elite, junior players is important given the high participation rate of youth in basketball across many countries [15] and the subsequent growing need to ensure data are available for the development of age- and sex-specific training and player management strategies in the sport. Furthermore, no study has compared the EL during match-play between male and female basketball players using LPS devices and accelerometry. Studies in other sports advocate the use of these technologies given they can provide a more comprehensive selection of variables that are easier to collect and process than those provided using time-motion analyses [16]. Specifically, LPS and accelerometers can provide similar measures to time-motion analyses such as distance covered and running speed [17,18,19], as well as other variables such as accelerations, decelerations, and mechanical loading, which are useful in quantifying EL in basketball players [4,20,21,22,23]. Consequently, a wider analysis beyond video-based time-motion analyses alone may provide more detailed insight into the EL imposed on elite, junior basketball players during competition according to sex and playing position. Therefore, the aim of this study was to compare the EL during competition between elite, junior male and female players using modern technologies. It was hypothesized that elite, junior, male basketball players will undergo more activity, including more intense actions, than elite, junior, female basketball players.

## 2. Methods

### 2.1. Subjects

Junior, male basketball players from 2 teams (n = 25; age: 17 ± 1 years; height: 1.93 ± 0.09 m; body mass: 85.2 ± 9.5 kg; guards, n = 12; forwards, n = 7; centres, n = 6) and junior, female basketball players from 4 teams (n = 48; age: 17 ± 1 years; height: 1.76 ± 0.07 m; body mass: 67.2 ± 6.2 kg; guards, n = 22 guards; forwards, n = 13; centres, n = 13) volunteered to participate in this study. Players were classified by coaches as guards (point guards and shooting guards), forwards (small forwards and power forwards) and centres. All players were competing in an elite, junior competition, the Madrid Junior Basketball Final Four, which is Madrid’s State Tournament, a stage before the Spanish national tournament (one of the teams analyzed ended as the champion of the Spanish Junior National Championship, and two of the teams analyzed advanced to the final stage of the Junior Euroleague Tournament). All players and their parents or legal guardians were informed of the aims, risks, and benefits of the study before signing written consent to allow the collection of data for scientific purposes. The local institutional human research ethics committee approved the study protocol (CIPI/18/195).

### 2.2. Observation Period

The competition was played during four consecutive days in the same arena with 4 male teams playing each of the other male teams and 4 female teams playing each of the other female teams. All players in this study performed approximately 10 h of team training and 6 h of gym-based conditioning per week during the season leading into competition. The schedule of the competition for each team is shown in Table 1. Each match consisted of 4 × 10-min quarters with 1 min separating each quarter and 15 min separating each half (i.e., between quarters 2 and 3). At least 10 min of actual playing time (while the match clock was running) had to be completed in the match being analyzed for player data to be included in the final sample for analysis. Consequently, 55 individual male match samples and 133 individual female match samples were included in the final analyses.

### 2.3. Procedures

EL was monitored using WIMU PRO devices (Realtrack Systems SL; Almería, Spain). These devices each include an accelerometer, a gyroscope and a magnetometer sampling at 100 Hz, and were attached to the upper back of players during matches with an adjustable harness. The system also uses 6 portable ultrawideband (UWB) antennae placed within 5 m of each corner and middle line of the court, collecting positioning data at 20 Hz. The system operates using triangulations between the antennae and the units every 50 ms. The time required to receive the signal is calculated by the device and the unit position (X and Y) is derived using one of the antennae as a reference. The antennae remained in the same position across the entire observation period to ensure consistency in the acquired data. The data were analyzed using the WIMU software (Realtrack Systems; Almería, Spain).

The variables used to indicate EL were (a) total distance covered (m), (b) high-intensity (14–21 km·h^−1^) distance covered (m), (c) sprint (>21 km·h^−1^) distance covered (m), (d) total accelerations (n), (e) total decelerations (n), (f) relative accelerations (n·min^−1^), (g) relative decelerations (n·min^−1^), (h) ratio of accelerations:decelerations, (i) player load (arbitrary units [AU]) calculated using the following equation: player load_n_ = √[(ACx_n_ – Acx_n-1_)^2^ + (ACy_n_ – Acy_n-1_)^2^ + (ACz_n_ – ACz_n-1_)^2^]/100, where AC_(X,Y,Z)_ = AC_Body (acceleration minus gravity), AC_Y_ is the lateral–medial axis acceleration, AC_X_ is the vertical axis acceleration and ACz is the anteroposterior axis acceleration, and (j) relative player load (AU·min^−1^).

The WIMU PRO has shown adequate reliability to measure team sport-specific movements [24,25]. Specifically, the UWB technology showed better accuracy (bias: 0.57–5.85%), test-retest reliability (% technical error of measurement [TEM]: 1.19), and inter-unit reliability (bias: 0.18%) in determining distance covered than global positioning system (GPS) technology (bias: 0.69–6.05%; %TEM: 1.47; bias: 0.25%) during intermittent, team sport activity [25]. Also, UWB showed better results (bias: 0.09%; intraclass correlation [ICC]: 0.979; bias: 0.01%) in measuring mean movement velocity than GPS technology (bias: 0.18%; ICC: 0.951; bias: 0.03%) during walking (<6 km·h^−1^) and running (>16 km·h^−1^) [25]. The accuracy of the UWB technology has also been tested indoors, showing high sensitivity to relative positioning on the court [26].

### 2.4. Statistical Analysis

Prior to the descriptive analysis and the comparisons of EL between sexes, the Shapiro-Wilk test confirmed all variables showed non-normal distributions (*p* < 0.05). Consequently, Mann–Whitney U tests (using the Montecarlo’s exact test with an alpha level of 99%) were used to analyze differences in EL between sexes. The test was run for all players together and separately for each playing position. To check the magnitude of the effect for each Mann–Whitney U test, the r effect size (r= Z/z/√n) was calculated and interpreted as small (0–0.30), moderate (0.31–0.50), or large (>0.50) in magnitude [27]. All analyses were performed using IBM SPSS Statistics for Windows (Version 23.0. IBM Corp, Armonk, NY, USA).

## 3. Results

The EL variables recorded during competition for elite, junior male and female players are presented in Table 2. Analyses revealed males covered more (*p* < 0.05) high-intensity and sprint distances, and completed a greater quantity (*p* < 0.05) of decelerations than female players. In contrast, female players experienced a greater (*p* < 0.05) ratio of accelerations:decelerations than male players.

The EL variables registered during competition relative to sex and playing position are presented in Table 3. Larger (*p* < 0.05) high-intensity and sprint distances were completed by males in all playing positions (guards, forwards, and centres) and greater (*p* < 0.05) decelerations were observed for males in the guard position compared to female players. Greater (*p* < 0.05) acelerations·min^−1^ were apparent for females in the forward position compared to male players. In addition, larger (*p* < 0.05) ratios of accelerations:decelerations were observed for female players in guard and centre positions than male players in these positions.

## 4. Discussion

This is the first study to compare EL between sexes in junior basketball players during an official tournament. Our results indicate elite, junior, male players covered significantly more high-intensity and sprint distances and performed significantly more decelerations during competition than female players, while females underwent a significantly greater acceleration:deceleration ratio. Despite these findings being consistent across all playing positions, some position-specific variations were apparent with male guards decelerating more than female guards and female forwards undergoing more relative (·min^−1^) accelerations than male forwards.

Intense match demands are of interest for team sport practitioners given they can contribute to injury [28] and inform the development of match-specific training drills [29] to optimally prepare players for competition. In this regard, the differences in high-intensity and sprint distances we observed across sexes are of particularly interest and might be related to physiological, tactical, and scheduling factors. Specifically, males have been shown to possess greater accelerative ability and sprinting speeds than females in a range of team sports [30,31,32], underpinned by higher levels of strength and power [33,34]. Indeed, sex differences in knee extensor and flexor strength increase with age from childhood to young adulthood, with the most prominent strength increases in male basketball players compared to female players evident in older age groups concomitant with our sample of players (16–22 years) compared to younger age groups (9–13 years) [34]. In turn, given standardized speed zones, rather than individualized speed zones relative to maximum capacities in each player were used in our study, the greater running speeds typical of male basketball players [32,35] would more frequently register in the high-intensity and sprint zones. Male basketball competition in the present study may also have elicited a faster pace of play than female match-play due to the tactical strategies adopted. In this sense, adult, male basketball competition has been shown to involve greater dribbling activity [11] than female competition, which may underpin more dribble penetration situations leading to a faster style of play [36]. In addition, the different scheduling of matches according to sex with males competing in the afternoon and females competing in the morning may have also contributed to variations in high-intensity and sprint activity between sexes. Indeed, sleeping behaviors may have been impacted to a greater extent in female players prior to morning competition compared to the male players competing later in the day, with basketball performance also showing diurnal fluctuations in favor of competing later in the day [37]. Nevertheless, sleeping behavior was not measured in this study and these observations remain speculative in nature.

Our results contrast the only previous study directly comparing EL between sexes during basketball match-play [11]. Specifically, Scanlan et al., [11] showed sprinting requirements were consistent across sexes when playing time was accounted for. Methodological differences between studies may account for variations in findings. First, Scanlan et al. [11] investigated semi-professional, adult, male and female players, while our study investigated junior players. In turn, less discrepancy between sexes may be apparent in match activities performed at high intensities following the adolescence period. Second, different criteria were used to classify sprinting across studies, with a higher speed threshold used previously (>25.2 km·h^−1^) than in our study (>21 km·h^−1^). Consequently, all activity registering between 10.8–25.2 km·h^−1^ were recorded as “running” by Scanlan et al. [11], which likely encompassed a wide range of activities and was possibly less sensitive in detecting differences in EL intensities between sexes than our approach. Third, variations in approaches to quantify EL were also apparent across studies, with earlier research (11), using video-based time-motion analyses across a smaller sample of matches (n = 6) than in our study (n = 11). In contrast, our study adopted the use of LPS and microsensors, which may possess varying levels of accuracy when quantifying EL [38]. Use of LPS and microsensor technologies may have some advantages over the time-motion analyses, such as a higher accuracy measuring low-intensity activities that involve changes in direction when performed in congested spaces [39], and more efficient data processing through proprietary software [16].

In addition to high-intensity and sprinting differences between sexes, we observed a lower quantity of decelerations in female players, which may also be linked with strength-related mechanisms. Given eccentric strength directly underpins decelerations and changes in direction [40], the lower eccentric strength documented in junior, female basketball players compared to age-matched male players [33] might be an important determining factor underpinning variations in match deceleration demands across sexes. Furthermore, the greater high-intensity and sprinting activity performed by male players during match-play likely placed an increased emphasis on decelerating to stop or change direction following these intense movement sequences. It should be noted, this study is the first to compare acceleration and deceleration demands between sexes in basketball and thus comparisons with past literature are difficult to make. Consequently, this novel outcome suggests greater emphasis on braking or initiating changes in direction may be placed on elite, junior, male players compared to their female counterparts during match-play.

While significantly greater decelerations were performed by male players overall in the present study, when analyzed further according to playing position, significant differences in deceleration demands were only evident in the guard position. This position may particularly be prone to sex differences in deceleration demands given female guards have been shown to perform less shuffling activity than male guards during match-play [11]. In turn, shuffling activity likely contributes extensive whole-body deceleration motions given acceleratory movements are made to initiate shuffling in a lateral direction with subsequent decelerations following planting of the leading foot underscored by simultaneous flexion of the ankle, knee, and hip and eccentric muscle action in a cyclic manner [41]. In addition, female players in the forward position performed more accelerations·min^−1^ than male players. Indeed, forwards have been shown to undergo heightened workloads compared to other playing positions in female basketball match-play [42], with the authors attributing these positional variations to forwards travelling further down the sidelines rather than the middle of the court during transitions across the court. These position-specific requirements may have promoted heightened acceleration work in the forward position during female match-play compared to male match-play. Research comparing the EL between sexes in handball competition [31] identified turnovers as a cause of more frequent transitions in play during female competition, which could translate to basketball given Scanlan et al. [11], suggested a greater number of transitions could result in shorter possession times during female match-play. In these turnover scenarios, forwards may be particularly prone to execute more accelerations than other playing positions given their sprinting capabilities and high mobility on the court compared to other playing positions [43].

While this study provides novel outcomes regarding the EL experienced during elite, junior basketball match-play according to sex, there were some limitations encountered. First, the present study investigated players during a tournament. In this regard, only 3 matches were analyzed per team, which is less than that experienced across a typical season in most competitions. Furthermore, while the competitive restraints in tournament scenarios were relatively consistent across teams, the influence of fatigue when playing on consecutive days might have impacted players differently and influenced results, particularly for playing positions typically possessing lower fitness (e.g., centres) [44]. Consequently, future studies should explore the EL during matches within tournament and regular (weekly competition) seasonal formats to identify key differences between sexes. Second, although a multi-team approach is a strength of the present study, the present results are limited to the teams investigated and may not be indicative of teams competing in other competitions. More work is encouraged exploring differences in EL during match-play between sexes in junior basketball players at other playing levels and in other competitions around the world. Third, this study was focused on EL, therefore the responses of players during matches was not reported. Internal loading (IL) is important as it indicates the physiological and perceptual stress encountered by players and therefore may help coaches to develop substitution and recovery strategies for players during and following matches [6]. Future research should explore EL and IL to provide comprehensive data regarding differences in match demands between sexes in elite, junior basketball players.

## 5. Conclusions

This is the first study to compare the EL encountered during elite, junior, male and female basketball competition. Results suggest male junior players cover greater distances at higher speeds than female players. Furthermore, males appear to perform more decelerations than female players, predisposing to a greater acceleration:deceleration ratio in female players. In addition, position-specific analyses mirrored the high-intensity and sprint differences between sexes evident for the whole sample, while greater decelerations were observed for male guards compared to females and higher acelerations·min^−1^ were apparent for female forwards compared to males.

The current findings may assist basketball coaches and conditioning professionals to optimize training processes. First, given the greater running intensities across positions in elite, male, junior players than female players, conditioning programs should be adapted according to sex as males may necessitate training drills that promote further distances running and sprinting than females to prepare for competition. Second, as female guards and centres performed a higher acceleration:deceleration ratio than males, training protocols including overall and specifically eccentric strength could be crucial for female players in these playing positions to decelerate effectively during match scenarios, and subsequently prevent injuries induced by possible imbalances in acceleratory demands [45]. Third, due to the higher decelerations performed by male guards compared to females, and the greater accelerations·min^−1^ performed by female forwards compared to males, position-specific conditioning drills should focus on acceleration–deceleration work for players occupying guard positions in male teams and forward positions in female teams to prepare for competition demands.

## Figures and Tables

**Table 1 ijerph-17-01456-t001:** Match schedule of the tournament.

Time (hh:mm)	Day 1	Day 2	Day 3	Day 4
10:30			female 2 vs 4	female 1 vs 4
12:30			female 1 vs 3	female 2 vs 3
18:30	female 1 vs 2	male 1 vs 2	male 2 vs 4	male 1 vs 4
20:30	female 3 vs 4	male 3 vs 4	male 1 vs 3	male 2 vs 3

Note: Teams 1 and 2 were not included in the analysis of the male competition.

**Table 2 ijerph-17-01456-t002:** The external loads (mean ± standard deviation) encountered by elite, junior male and female players during competition.

Variable	Male (n = 50)	Female (n = 113)	Statistical Outcomes			
			*Z*	*p*	*r*	*Effect*
Total distance (m)	2776 ± 1529	2513 ± 1300	−0.788	0.431	−0.06	Small
High-intensity distance (m)	453 ± 263	237 ± 170	−5.117	<0.001 **	−0.39	Moderate
Sprint distance (m)	49 ± 59	14 ± 24	−6.148	<0.001 **	−0.46	Moderate
Accelerations (n)	372 ± 273	370 ± 285	−0.322	0.747	−0.02	Small
Decelerations (n)	367 ± 273	273 ± 239	−2.447	0.014 *	−0.18	Small
Relative accelerations (n·min^−1^)	8.6 ± 5.2	9.1 ± 5.3	−0.993	0.321	−0.08	Small
Relative decelerations (n·min^−1^)	8.5 ± 5.2	6.5 ± 3.7	−1.375	0.169	−0.10	Small
Accelerations:Decelerations	1.02 ± 0.04	1.61 ± 0.79	−3.552	<0.001 **	−0.27	Small
Player load (AU)	43 ± 27	39 ± 21	−0.399	0.690	−0.03	Small
Relative player load (AU·min^−1^)	0.95 ± 0.37	0.98 ± 0.35	−0.752	0.452	−0.06	Small

Note: AU = arbitrary units; * *p* < 0.05; ** *p* < 0.01.

**Table 3 ijerph-17-01456-t003:** The external loads (mean ± standard deviation) encountered by elite, junior male and female players according to playing position during competition.

Variable	Male	Female	Statistical Outcomes			
			*Z*	*p*	*r*	*Effect*
**Guards**						
Sample size (n)	19	52				
Total distance (m)	2963 ± 1811	2175 ± 1227	−1.701	0.089	−0.18	Small
High-intensity distance (m)	443 ± 303	191 ± 145	−3.286	0.001 **	−0.34	Moderate
Sprint distance (m)	53 ± 71	13 ± 22	−3.915	<0.001 **	−0.41	Moderate
Accelerations (n)	405 ± 320	331 ± 256	−0.779	0.436	−0.08	Small
Decelerations (n)	401 ± 320	230 ± 199	−2.377	0.017 *	−0.25	Small
Relative accelerations (n·min^−1^)	8.6 ± 5.6	8.8 ± 5.4	−0.169	0.866	−0.02	Small
Relative decelerations (n·min^−1^)	8.5 ± 5.6	5.9 ± 3.4	−1.546	0.122	−0.16	Small
Accelerations:Decelerations	1.02 ± 0.04	1.61 ± 0.78	−2.169	0.030 *	−0.22	Small
Player load (AU)	45 ± 31	35 ± 21	−1.039	0.299	−0.11	Small
Relative player load (AU·min^−1^)	0.90 ± 0.40	0.93 ± 0.37	−0.286	0.775	−0.03	Small
**Forwards**						
Sample size (n)	15	30				
Total distance (m)	3050 ± 1523	3432 ± 1193	−1.035	0.301	−0.14	Moderate
High-intensity distance (m)	515 ± 256	324 ± 205	−2.022	0.043 *	−0.27	Small
Sprint distance (m)	44 ± 29	15 ± 19	−3.493	<0.001 **	−0.47	Moderate
Accelerations (n)	381 ± 249	497 ± 305	−1.228	0.219	−0.17	Small
Decelerations (n)	376 ± 250	392 ± 264	−0.169	0.866	−0.02	Small
Relative accelerations (n·min^−1^)	8.3 ± 4.8	10.0 ± 5	−2.071	0.038 *	−0.28	Small
Relative decelerations (n·min^−1^)	8.2 ± 4.8	8.0 ± 4.2	−0.746	0.455	−0.10	Small
Accelerations:Decelerations	1.03 ± 0.05	1.64 ± 0.81	−1.806	0.071	−0.24	Small
Player load (AU)	47 ± 30	52 ± 20	−1.035	0.301	−0.14	Small
Relative player load (AU·min^−1^)	0.97 ± 0.37	1.12 ± 0.27	−1.685	0.092	−0.23	Small
**Centres**						
Sample size (n)	16	31				
Total distance (m)	2296 ± 1081	2193 ± 1106	−0.269	0.788	−0.04	Small
High-intensity distance (m)	405 ± 220	230 ± 143	−2.784	0.005 *	−0.37	Moderate
Sprint distance (m)	47 ± 67	16 ± 31	−3.041	0.002 *	−0.40	Moderate
Accelerations (n)	324 ± 243	313 ± 283	−0.741	0.459	−0.10	Small
Decelerations (n)	320 ± 242	231 ± 244	−1.628	0.104	−0.22	Small
Relative accelerations (n·min^−1^)	8.8 ± 5.5	8.3 ± 5.0	−0.022	0.982	0.00	Small
Relative decelerations (n·min^−1^)	8.7 ± 5.5	6.1 ± 3.5	−0.988	0.323	−0.13	Small
Accelerations:Decelerations	1.01 ± 0.05	1.57 ± 0.83	−2.190	0.029 *	−0.29	Small
Player load (AU)	37 ± 20	34 ± 19	−0.404	0.686	−0.05	Small
Relative player load (AU·min^−1^)	1.00 ± 0.35	0.94 ± 0.36	−0.202	0.840	−0.03	Small

Note: AU = arbitrary units; * *p* < 0.05; ** *p* < 0.01.

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
