# Peer review of "Comparing the External Loads Encountered during Competition between Elite, Junior Male and Female Basketball Players"

_ijerph, 2020, doi:10.3390/ijerph17041456_

Round 1
Reviewer 1 Report
Dear authors,
Thank you so much to allow me review the manuscript entitled "Comparing the external loads encountered during competition between elite, junior male and female basketball players". I enjoyed reading it and I think it is very well written and has a great impact on basketball training. However, there are some small concerns that must be resolved:
- Several marks of the track-change are observed throughout the manuscript, which must be removed.
L15. It is true that 11 matches were analyzed, but I think it is necessary to specify that 3 matches/team were analyzed. This must be also stated in the methods section.
L26-28. Please, provide a more specific practical application.
-The introduction section is very well written. Congratulations.
- Given that the general dynamics of basketball comprise one game per week and here a congested period of competition is valued, I consider that it should be refelected.
L69. It would be advisable to refer to the comparison between playing positions.
L71. Could you include a hypothesis?
Methods
L81-82. Since the players were minors, it is necessary that the informed consent be signed by their parents or legal guardians.
Results
Could the authors clarify the reason why some values are presented as absolute values and others as relative values? I believe that all values should be presented as relative, since not all players participate in the same number of minutes in each match. If the authors disagree, please explain and justify your opinion.
Discussion
- Very interesting section.
L154-155. Please, state "during an official tournament"
L235-240. I believe that this is not a limitation, but that information about the reality of basketball is provided. I think a limitation is the number of matches analyzed per team (3).
Author Response
Reviewer 1 (Red Corrections)
Abstract
L15. It is true that 11 matches were analyzed, but I think it is necessary to specify that 3 matches/team were analyzed. This must be also stated in the methods section.
We agree with the observation and proceeded to change it in the manuscript.
L26-28. Please, provide a more specific practical application.
Given the limited space available in the abstract, we tried to be concise in providing a practical recommendation sentence. However, we can see potential to be more specific in our recommendation and have modified the text to now read as: “These outcomes can be used in developing sex- and position-specific training plans, and in turn improving the physical preparedness of junior basketball players for competition demands at the elite level.” We hope that you can find it more specific.
Introduction
L69. It would be advisable to refer to the comparison between playing positions.
We agree and proceeded changed it in the text so that “playing position” is mentioned.
L71. Could you include a hypothesis?
Thank you for this suggestion, we have now included a hypothesis at the end of the introduction as requested.
Methods
L81-82. Since the players were minors, it is necessary that the informed consent be signed by their parents or legal guardians.
We acknowledge the need for consent from parents or legal guardians when researching minors as subjects. We apologize for omitting this information in our original submission, but we can confirm parental/guardian consent was obtained for each subject prior to study commencement. This detail has now been added to the manuscript.
Results
Could the authors clarify the reason why some values are presented as absolute values and others as relative values? I believe that all values should be presented as relative, since not all players participate in the same number of minutes in each match. If the authors disagree, please explain and justify your opinion .
Thanks for your comment. The main variables were presented as absolute values, and only some specific ones related to relative ratios were specified to improve the meaning of accelerations, decelerations and player load. This approach allowed a better understanding when comparing the variables studied with previous research focused on physical demands in basketball.
Discussion
L154-155. Please, state "during an official tournament"
The manuscript text has been changed as requested.
L235-240. I believe that this is not a limitation, but that information about the reality of basketball is provided. I think a limitation is the number of matches analyzed per team (3).
Thank you for providing this insight. In addressing examination of a tournament scenario compared to a regular season as a limitation, we have made emphasis on the limited number of matches analyzed per team as a limitation in the manuscript to acknowledge this point.

Reviewer 2 Report
I have a serious concern regarding human ethics reported in this investigation.
Bearing in mind that the study populations are comprised of minors (chronological age average is 16 on average) they should NOT have signed informed consent.
Rather, their parents should. In case this is not true, this research paper should be withdrawn.
This seems to be a classical violation of the Helsinki (1964) rules and regulations
Author Response
“I have a serious concern regarding human ethics reported in this investigation.
Bearing in mind that the study populations are comprised of minors (chronological age average is 16 on average) they should NOT have signed informed consent.
Rather, their parents should. In case this is not true, this research paper should be withdrawn.
This seems to be a classical violation of the Helsinki (1964) rules and regulations”
We acknowledge the need for consent from parents or legal guardians when researching minors as subjects. We apologize for omitting this information in our original submission, but we can confirm parental/guardian consent was obtained for each subject prior to study commencement. This detail has now been added to the manuscript.

Reviewer 3 Report
General comments
The authors present the external load experienced by players in a junior basketball tournament scenario and compare variables between positions and sexes. The article is well written and provides useable inferences for coaches and practitioners working with youth basketball players. I do, however have some minor concerns that I believe should be address before this article can be accepted for publication.
Title
After reading the article I do not believe the sample here can be classified as “elite”, I feel this needs to be changed in the title.
Abstract
Generally well written and provides a good overview of the study
Line 16, I would just detail sprint thresholds as >21 km/h, no one is actually going to run anywhere near 50 km/h.
Line 23, spelling mistake in accelerations
Introduction
Well written with appropriate information included that sets up the research question well, very few comments here.
Line 34, I think it would be a good idea to clarify what is meant by internal characteristics here.
Line 63, I would re word “more comprehensive suite of variables”, the current wording is somewhat strange in my opinion.
Methods
Line 24, no decimal places needed for age, all values should be reported to the number of decimal places to which they are recorded. Please amend throughout.
Line 79, what exactly classifies this competition as “elite”? I am generally do not like the use of this word unless it can be justified.
Line 81, if this below national level, I do not consider this elite and I’m sure many others would agree.
Line 108, as stated in my comments re the abstract; I would just detail sprint thresholds as >21 km/h, no one is actually going to run anywhere near 50 km/h.
Stats, could you please clarify if the statistical analyses employed here account for uneven samples.
Results
Easy to interpret, with well presented tables.
Line 138, I would change alternatively to in contrast.
Discussion
Line 167, power is a very vague term which is often used incorrect, I would clarify exactly what you mean here.
Line 177-183, I feel this explanation around sleep is somewhat speculative, I would maybe include a sentence at the end stating what sleep was not measured here and this suggestion remains speculative.
Line 191, please clarify why different speed threshold were selected in the present study,
The limitations of the study are well acknowledged.
Line 261, As previously stated I do not feel that this populations can be classified as “elite”.
Author Response
Reviewer 3 (Blue Corrections)
Abstract
Generally well written and provides a good overview of the study
Line 16, I would just detail sprint thresholds as >21 km/h, no one is actually going to run anywhere near 50 km/h.
21-50km/h is the threshold that the manufacturer has set by default in the devices, that is why we reflected it that way. But the request makes sense so we changed it in the text for ease of interpretation.
Line 23, spelling mistake in accelerations
Thank you for identifying this error, it is been corrected in the manuscript.
Introduction
Well written with appropriate information included that sets up the research question well, very few comments here.
Line 34, I think it would be a good idea to clarify what is meant by internal characteristics here.
Thank you for pointing out the lack of clarity in our writing. We concede that “characteristics” is probably not the best word to use in this instance, and have since changed it to “responses”. Furthermore, we have included examples of internal responses (heart rate, rating of perceived exertion) to help readers understand the terminology better.
Line 63, I would re word “more comprehensive suite of variables”, the current wording is somewhat strange in my opinion.
We have amended this text in the manuscript to now read more clearly as: “…more comprehensive selection of variables that are easier to collect and process than those provided using time-motion analyses.”
Methods
Line 24, no decimal places needed for age, all values should be reported to the number of decimal places to which they are recorded. Please amend throughout.
Thank you for highlighting this error. We have amended the number of decimal places used in reporting data, including age, as appropriate throughout the manuscript.
Line 79, what exactly classifies this competition as “elite”? I am generally do not like the use of this word unless it can be justified.
We can recognize the difficulties in interpreting the descriptor “elite” in studies, so to rectify this issue, we have now included further explanation of the playing level in the manuscript. Specifically, while we identified the competition as the Madrid Junior Basketball Final Four, which is Madrid´s State Tournament, a stage before the Spanish national tournament, we have clarified that two of the teams competing in this tournament classified for the final stage of the Junior Euroleague basketball competition and one of them was the Spanish national champion to further demonstrate the high (elite) level of the teams investigated. There were also several international players competing in the tournament. Currently four of the players are already professionals in top European teams.We feel that this information provides sufficient context to what we considered as “elite” in our study.
Line 81, if this below national level, I do not consider this elite and I’m sure many others would agree.
Explained above
Line 108, as stated in my comments re the abstract; I would just detail sprint thresholds as >21 km/h, no one is actually going to run anywhere near 50 km/h.
Thank you again for this suggestion, we have amended the text in the manuscript to address this point as in the Abstract.
Stats, could you please clarify if the statistical analyses employed here account for uneven samples.
Thanks for your comment, due to the nature of data (non-parametric distribution and that the two population variances are not assumed to be equal) the non-parametric tests were used.
Results
Easy to interpret, with well presented tables.
Line 138, I would change alternatively to in contrast.
It has been changed in the manuscript as suggested.
Discussion
Line 167, power is a very vague term which is often used incorrect, I would clarify exactly what you mean here.
We acknowledge that power can encompass many different measures and is sometimes used incorrectly, so to clarify this point, we have indicated what measures were used to indicate power in the studies cited after the sentence. Please keep in mind that this sentence aims to provide evidence that strength and power are higher in men than in women in team sports.
Line 177-183, I feel this explanation around sleep is somewhat speculative, I would maybe include a sentence at the end stating what sleep was not measured here and this suggestion remains speculative.
We agree with your suggestion to acknowledge the speculative nature of our reasoning put forward here and have added the requested sentence in the manuscript.
Line 191, please clarify why different speed threshold were selected in the present study,
To better clarify why we adopted different speed thresholds, the following amendments were made to the manuscript: “Consequently, all activity registering between 10.8–25.2 km·h-1 were recorded as “running” by Scanlan et al. [11], which likely encompassed a wide range of activities and was possibly less sensitive in detecting EL differences between sexes according to intensity than our approach.” We feel it is implicitly stated why we added different speed thresholds in this sentence in that it was to ensure more sensitive to detection of differences in external load intensities could be made between sexes.
The limitations of the study are well acknowledged.
Line 261, As previously stated I do not feel that this populations can be classified as “elite”.
Explained above. Let we know if you need more details.

Round 2
Reviewer 2 Report
I am still not convinced, I believe that you had overlooked the Ethical issues. Thus I must decline the acceptance of the present MS.